# Decision to self-isolate during the COVID-19 pandemic in the UK: a rapid scoping review

Claire Marriott Keene [1], Sophie Dickinson,[2] Reshania Naidoo [1,2]
Billie Andersen-Waine,[2] Angus Ferguson-Lewis,[2] Anastasia Polner,[2]
Ma'ayan Amswych,[2] Lisa White,[3] Sassy Molyneux,[1,4] Marta Wanat [5] EY-Oxford
Health Analytics Consortium

[1]Centre for Global Health Research, Nuffield Department of Medicine, University of Oxford, Oxford, UK
[2]UKI Health Sciences and Wellness, Ernst & Young (EY), London, UK
[3]Department of Biology, University of Oxford, Oxford, UK
[4]Pandemic Sciences Institute, University of Oxford, Oxford, UK
[5]Nuffield Department of Primary Care Health Sciences, University of Oxford, Oxford, UK

**Correspondence to**
Dr Claire Marriott Keene;
clairekeene@gmail.com

## ABSTRACT

**Objective** Testing for COVID-19 was a key component of the UK's response to the COVID-19 pandemic. This strategy relied on positive individuals self-isolating to reduce transmission, making isolation the lynchpin in the public health approach. Therefore, we scoped evidence to systematically identify and categorise barriers and facilitators to compliance with self-isolation guidance during the COVID-19 pandemic in the UK, to inform public health strategies in future pandemics.

**Design** A rapid scoping review was conducted.

**Search strategy** Key terms were used to search literature databases (PubMed, Scopus and the WHO COVID-19 Research Database, on 7 November 2022), Google Scholar and stakeholder-identified manuscripts, ultimately including evidence published in English from UK-based studies conducted between 2020 and 2022.

**Data extraction and synthesis** Data were extracted and synthesised into themes, organised broadly into capability, opportunity and motivation, and reviewed with key stakeholders from the UK Health Security Agency (UKHSA).

**Results** We included 105 sources, with 63 identified from UKHSA and used to inform their decision-making during the pandemic. Influences on the decision to comply with isolation guidance were categorised into six themes: perceived ability to isolate; information and guidance; logistics; social influences, including trust; perceived value; and perceived consequences. Individuals continuously assessed these factors in deciding whether or not to comply with guidance and self-isolate.

**Conclusions** Decisions to self-isolate after a positive test were influenced by multiple factors, including individuals' beliefs, concerns, priorities and personal circumstances. Future testing strategies must facilitate meaningful financial, practical and mental health support to allow individuals to overcome the perceived and actual negative consequences of isolating. Clear, consistent communication of the purpose and procedures of isolating will also be critical to support compliance with self-isolation guidance, and should leverage people's perceived value in protecting others. Building public trust is also essential, but requires investment before the next pandemic starts.

## STRENGTHS AND LIMITATIONS OF THIS STUDY

⇒ This review is strengthened through the use of a broad search strategy to identify a wide range of evidence and by engagement with a diverse group of stakeholders to contextualise and accurately interpret the results.

⇒ It is also strengthened through the inclusion of a large body of unpublished and confidential internal data from the UK Health Security Agency, which supplemented the limited published literature and which would not otherwise be available for synthesis, general dissemination and contribution to discussions on future pandemic strategies.

⇒ However, given the paucity of studies specifically addressing isolation behaviours, much of the published evidence on isolation behaviour in this review was drawn from studies that focused primarily on testing rather than isolation during the COVID-19 pandemic.

⇒ Additionally, the review was conducted over a short period of time by a large team, potentially introducing inconsistencies into the screening and extracting processes despite attempts to mitigate bias, including overlap of screening and extraction between reviewers, quality checks and review of included original documents during write up.

## INTRODUCTION

Any novel pandemic poses a particular challenge in that definitive pharmaceutical interventions, such as therapeutic medications or preventative vaccines, take time to develop.[1] In the meantime, public health policy is reliant on non-pharmaceutical interventions to reduce transmission and lessen impact.[1 2] In the UK, the initial response to COVID-19 centred on a spectrum of non-pharmaceutical public health measures aimed at mitigating transmission from COVID-19-positive cases[1 2]: periods of national and local lockdown, social distancing, limits on social gatherings, mandated use of face coverings in public

spaces and self-isolation after a positive COVID-19 test result or contact with a case.[3 4]

To facilitate isolation, mass testing for COVID-19 was required to identify COVID-19-positive individuals and their close contacts.[5] The National Health Service Test and Trace (NHSTT) programme was established in May 2020 to implement testing, tracing and isolation services at scale.[6] This complex intervention was the first programme of its kind in the UK, created and delivered at pace during a period of unprecedented uncertainty.[5] It distributed nearly 2 billion lateral flow tests and 158 million PCR tests free of charge in England alone, at a reported cost of £25.8 billion, with the aim of preventing population-wide deaths, decreasing morbidity, reducing burden on healthcare services and facilitating the reopening of the economy. The testing programme detected between 26% and 40% of all COVID-19 cases in England.[5]

The ultimate goal of the testing and tracing activities was to facilitate isolation of those who tested positive for COVID-19 and/or who were close contacts of positive cases; it was through the isolation step of the NHSTT programme's cascade that transmission was reduced.[7] Consequently, regardless of the resources invested in encouraging people to test, if people did not isolate when required, the impact of the programme would be undermined. Despite requirements to self-isolate, compliance was often suboptimal and varied across different contexts and over time.[5] It is crucial to understand what drove adherence with isolation guidance, and what barriers were faced, to inform future pandemic response strategies.

To support the development of more robust public health strategies, we conducted a scoping review of the available evidence on perceptions and experiences of self-isolation during the COVID-19 pandemic in the UK. We then categorised the identified barriers and facilitators to self-isolation.

## METHODS

### Study design

A rapid scoping study was conducted to summarise the large volume of rapidly generated, heterogenous evidence on isolation during the COVID-19 pandemic in the UK, to identify gaps in this evidence, and to describe the barriers and facilitators to engaging with self-isolation guidance. In conducting this scoping review, we followed the 2005 'Arksey and O'Malley' framework[8] (with the adaptations proposed by Levac *et al* in 2010[9]) and drew on the 2015 Joanna Briggs Institute guidance.[10]

This scoping study was nested in a broader rapid scoping review of barriers and facilitators to engaging with the COVID-19 testing programme, covering testing, reporting and self-isolation behaviour in the UK. This in turn formed part of a mixed-method programme of work conducted by the EY-Oxford Health Analytics Consortium, aimed at evaluating the UK Health Security Agency's (UKHSA's) COVID-19 testing strategy in England—the protocol and report are available online.[5 11]

### Search strategy and selection of the evidence

The search strategy (see online supplemental table 1 for search terms) was developed in close consultation with an information specialist and conducted on 7 November 2022 using PubMed, Scopus, the WHO COVID-19 Research Database and documents provided by UKHSA. UKHSA-provided documents comprise a body of published and unpublished research; many of these documents were confidential internal documents relating to work specifically conducted to inform UKHSA's decisions during the pandemic. This search was supplemented through free-text searches on Google Scholar, review of the references of included articles and stakeholder consultation[12] (see online supplemental table 2 for the rationale for database selection).

Following the database search, all identified citations were collated and uploaded into Rayyan (a web-based research collaboration platform[13]), and duplicates were removed. Following an initial screening pilot, titles and abstracts were screened by two reviewers against the inclusion criteria for the overall project, including testing, reporting and isolation behaviour. A sample of ≥20% titles and abstracts were evaluated by a third reviewer to clarify eligibility criteria and ensure consistency of inclusion,[9] calculating a Gwet's first-order coefficient (AC1)[14] to assess agreement. Potentially relevant sources were retrieved in full and reviewed against the eligibility criteria. Disagreements at each stage of the selection process were resolved through discussion; discussions were held with an additional reviewer if no consensus was reached. The evidence included in this paper on isolation behaviour was drawn from the documents included in the overarching review (table 1).

### Data extraction, charting and synthesis

To rapidly process a large volume of documents, a team of 12 reviewers extracted the data. Data extracted included (1) study metadata and (2) information about the perceptions, experiences, barriers and facilitators to each of the key activities (testing, reporting and isolating). Data were extracted into an Excel template, refined through entry of initial sources (online supplemental table 3). Each reviewer extracted data from two sources that overlapped with sources reviewed by another team member, to check quality and support discussions to refine the template and eligibility criteria. Given the rapid timelines for the overarching programme of work and the aim of this study to scope the evidence, the articles were not assessed for quality.

The analysis combined deductive and inductive elements, organised across three steps. First, a set of a priori categories was developed based on the expertise of the team and an initial reading of included sources. Initial categories of barriers and facilitators were refined throughout the analysis to ensure they reflected the full dataset. Second, the data were synthesised thematically, resulting in some categories being collapsed and reconfigured. Third, the themes were mapped onto the

**Table 1** Summary of the search parameters and limits as well as the inclusion and exclusion criteria, categorised according to the 'population, context, concept' search framework[103]

| | Inclusion criteria | Exclusion criteria |
|---|---|---|
| Search limits | | |
| Language | Published in English | Published in languages other than English |
| Dates | Published from 2020 | Published before 2020 |
| Methods | Qualitative or mixed methods studies | Purely quantitative studies |
| Eligibility | | |
| Literature | Journal articles, peer-reviewed material, articles under review, published books and book chapters, other academic research, research commissioned by governments, unpublished reports | Opinion or statement pieces, magazine articles, blog posts |
| Population | England, Northern Ireland, Scotland, Wales, and the islands making up the British Isles | Countries outside the UK, including the Ireland |
| Context (service settings) | All NHSTT programme service settings, including healthcare workers, adult social care, schools, universities, community, and the universal testing programmes. | Nil |
| Concept (key activities) | Description of isolation behaviour and barriers and/or facilitators to adherence to self-isolation guidance (with a focus on isolating due to a positive COVID-19 test result, but including isolating after exposure to a positive contact). | Not focused on the behaviour of self-isolation |

NHSTT, National Health Service Test and Trace.

dimensions of the Capability, Opportunity, and Motivation Model of Behaviour (COM-B)[15] (figure 1), to facilitate selection of potential intervention strategies.

### Stakeholder input

Stakeholders from UKHSA were consulted to identify additional sources of published and unpublished evidence (included as 'stakeholder-identified' sources), sense-check the findings and help interpret, frame and contextualise the results.

## RESULTS
### Overview of the evidence

After screening 4152 records, this analysis included 105 sources in the synthesis on self-isolation during COVID-19 (see figure 2, full details in online supplemental table 4). Reasons for exclusion at full-text screening are described in figure 2. Of these 105 studies, 29% were identified through the literature database search, 60% from UKHSA stakeholders, and 11% from other sources (bibliographic review and Google search). Overall, 81%

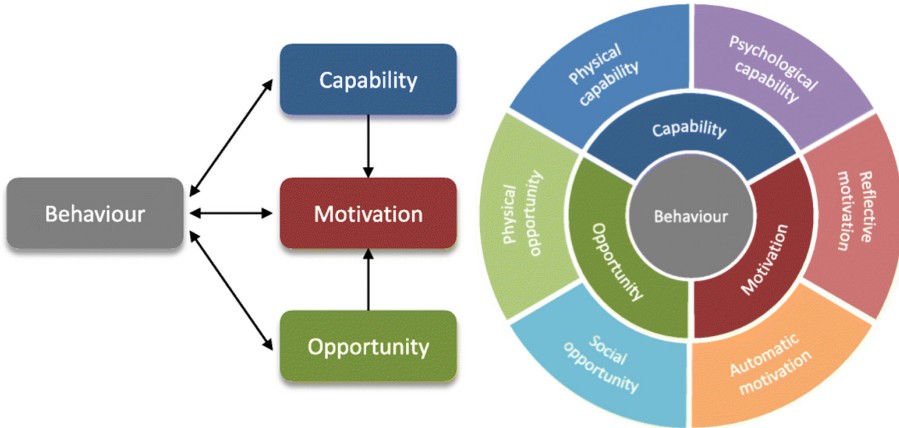

**Figure 1** The dimensions of the Capability, Opportunity, and Motivation Model of Behaviour, adapted from McDonagh *et al*'s application to a systematic review of chlamydia testing facilitators.[104]

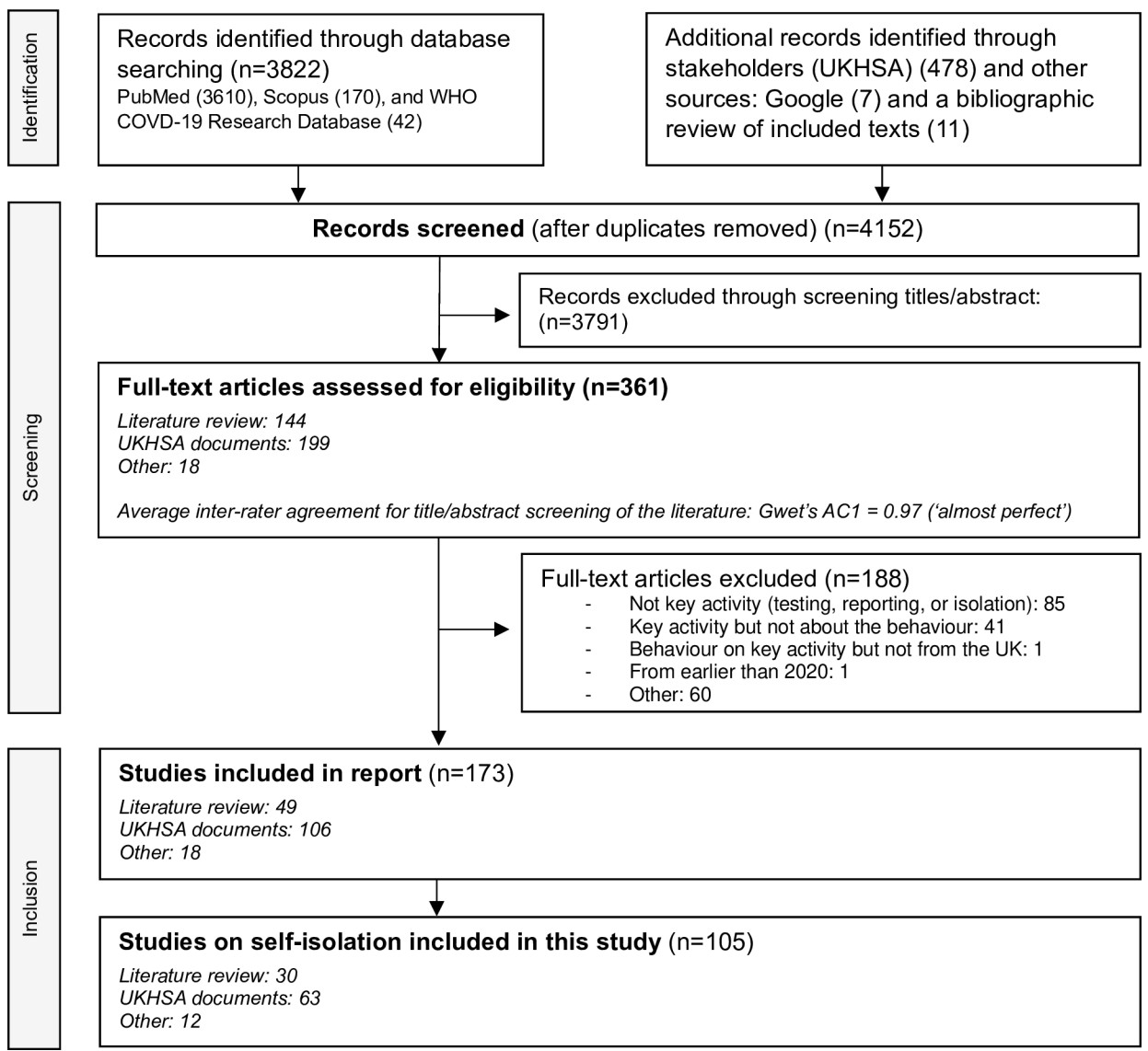

**Figure 2** Preferred Reporting Items for Systematic Reviews and Meta-Analyses extension for scoping review flow diagram outlining the search and inclusion of the literature. UKHSA, UK Health Security Agency.

were focused on England, 3% on Scotland, 2% on Wales and none on Northern Ireland; 12% covered the UK in general; and 2% included the UK in a broader international study. Data were collected during the early stages of the pandemic (before August 2020) in 18% of papers, during the middle stages (up to November 2021) in 75%, and during the final stages of restrictions in 18% (online supplemental table 5 and figure 1). Surveys were used in 37% of the sources, interviews in 54%, focus groups in 16% and other methods in 16%.

## Overview of results

We arrived at six themes to categorise the barriers and facilitators that influenced the decision to comply with isolation guidance (figure 3): logistics of isolation; social influences and trust; perceived ability to isolate; information and guidance; perceived consequences of complying with isolation; and perceived value of isolating. Ultimately,

individuals weighed up these influences in their decision to comply with guidance and self-isolate.

### Perceived ability to self-isolate

Despite negative descriptions of isolation, many people reported being able to self-isolate in line with contemporary requirements[16–19] and confidence in being able to self-isolate effectively and for the full duration required.[17] However, this varied among people and over time, as well as by the level of adherence to the guidance.[20]

Isolation was described as something that could be prepared for,[21 22] with previous experiences influencing whether and how individuals planned for potential periods of self-isolation.[22]

> I was aware that this could happen at any moment in time, more in actual fact during the winter time we had snow up here in the North of England [where] we got cut off for a few days, I've had flu before so I've

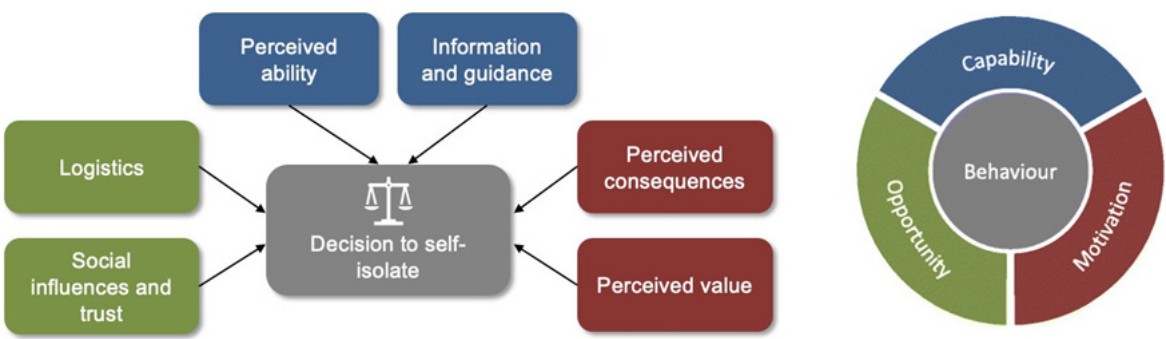

**Figure 3** Overview of the themes describing the influences on people's decision to comply with isolation guidance.

always got lots of soup, things in the freezer, it's kind of an eventuality so I didn't even have to ask somebody to go shopping for me, I just said right we're on lockdown until the test comes back so we had five or six days in the house where we just stayed at home. (Parent 4).[22]

External support reduced the need for people to leave home[21] and reduced reports of experiencing isolation as a challenge.[23] Required support included practical help, food, medical supplies, financial aid (including sick pay) and mental healthcare (such as welfare check-in calls or activities to distract those isolating).[24–28] Neighbourly support was also described as fostering a sense of community[28]; for some university students, the opportunity to isolate together in a single space facilitated mutual support and therefore facilitated self-isolation.[29]

Support came from the government through the Test and Protect system, the Test and Trace Support Payment Scheme,[30] local authority support measures such as shopping assistance,[31 32] the entitlement to sick pay[33] and universities.[28] Other support came through friends, family and community networks.[21 22 25 28] Support was described as particularly important for 'younger people, those with precarious incomes and women',[34] as well as people 'living alone or without existing social connections established in the community'.[35] Those with potentially greater need for support were less likely to report having adequate support.[23]

The uptake of existing support programmes was reduced by lack of awareness[30 36–38] and access issues,[30 36] including bureaucratic application processes and stringent eligibility criteria.[36 39] However, expanding eligibility alone had mixed effects on adherence to self-isolation.[39–41] It was suggested that support should also be widely advertised and proactively signposted through support call-handlers[36 42] and that reassurance was needed to overcome stigma and some people feeling unable to ask for help.[25]

### Information and guidance: understanding when, where, how and why to isolate

Poor understanding or confusion about the guidance and about transmission risks was a barrier to isolating effectively[20 24 43–46]; reported lack of clarity tended to be in understanding the rules about leaving home for further COVID-19 testing,[17] understanding the duration of isolation and when to start isolating,[43] understanding whether others in the household should isolate,[29] or whether to isolate if asymptomatic.[24 46] Understanding of guidance decreased when regulations changed.[47] One survey found greater understanding of the guidance among those who did self-isolate than among those who did not comply,[17] but this was not always consistent as experiences were also reported of isolating despite not understanding the guidance.[48 49]

Good communication was described as an enabler of adherence to isolation guidance,[39] including provision of written information in different languages and welfare check-in calls that increased knowledge of the guidance.[26] Communication of specific content was suggested to improve isolation uptake: providing insight into the importance of adherence to self-isolation guidance, demonstrating that it was the norm and that many people were following the guidance, reinforcing trust in the government and National Health Service (NHS) programmes, and advertising available practical and financial support.[36 50] On the other hand, uncoordinated and inefficient communication during isolation caused frustration.[25]

### Logistics of isolation: convenience and practical challenges

People described various practical challenges to following isolation guidance to the letter,[51] including limited access to testing,[24] unsafe or unsuitable home environments, managing childcare and caring responsibilities,[52] the difficulty of managing multiple positive cases or mixed results within a single household,[26 53] and avoiding other household members becoming sick.[22 43 51] These challenges were magnified in smaller houses and apartments[52] or when households included vulnerable family members.[29]

Ultimately, it appeared that combinations of these challenges heightened barriers to effective isolation.[26]

> I think it's unrealistic to isolate within the household. If my daughter gets it, it's unrealistic for her to stay in her room and not come out. (Parent).[43]

### Social influences and trust in authorities

People perceived high levels of adherence to self-isolation guidance in the community and higher still in their friends and family.[17] It was suggested that the perception that others were following self-isolation guidance, and that isolating when positive was the 'norm', facilitated self-isolation.[50]

Guidance from authoritative sources influenced adherence to self-isolation.[17] However, this deference to authority decreased over time,[54] and distrust in the government was described as a barrier to self-isolation.[24] Disagreement with government measures,[17] mistrust regarding the accuracy of the NHS COVID-19 app and proximity alerts,[22 52 55] and concern over secure handling of details by NHSTT seemed to be associated with lower adherence to self-isolation guidance.[17] A lack of trust in test results undermined engagement with isolation; the negative consequences of isolation (outlined below) were considered an unacceptable burden if the requirement to self-isolate might be 'unnecessary' if based on a false-positive result. This was a particular concern among healthcare workers[25] and care home staff.[56] Negative publicity may have undermined trust in testing processes and, in turn, willingness to isolate.[52]

> 'Several of those interviewed were concerned about reports of accuracy, concerning the LFDs [lateral flow devices]. There was a particular concern about false-positive tests and its potential to reduce their already fragile workforce in the event that they ended up having to send "non-infectious" employees home to self-isolate because of faulty test results.'[56]

### Perceived value in complying with isolation

Perceived benefit influenced the decision to comply with isolation guidance.[24] People described adherence to isolation guidance as 'important'[17] primarily to prevent transmission and protect others.[22 25 57] Adherence to self-isolation even beyond official requirements appeared to be driven by perceived risk of contracting COVID-19 and perceived vulnerability to severe COVID-19.[17 57] For these reasons, it was suggested that adherence to self-isolation would be supported through 'strengthening perceived benefit to self-isolate with messages emphasising its effectiveness'[24] and that the desire to protect others should be leveraged in communication of guidance.[45]

Some healthcare workers valued self-isolation as a break from heavy workloads[58] and an opportunity to 'engage in new hobbies, talk to family and friends by phone or online'[35] and foster a 'sense of community with others in the same situation'.[28]

In contrast, others felt that there was no benefit to isolating or following the rules, with some reporting it to be a 'waste of time',[59] particularly when they considered themselves low risk due to vaccination, had low-risk contacts, or used infection control measures.[60] This was particularly the case after a positive contact rather than after a positive test or if they had symptoms.[61–64] For some, perceived risk decreased over the course of the pandemic,[57] and students felt at lower risk because of their younger age.[28] Additionally, asymptomatic individuals who tested positive had lower motivation to isolate than symptomatic individuals testing positive, as observed by call handlers.[29]

### Perceived consequences of isolation
*Daily life*

People described the consequences of self-isolation on their daily life, such as the inability to leave one's house, socialise or use public transport.[65] This made it difficult to maintain routine domestic tasks (such as shopping, accessing healthcare, collecting medications or exercising), maintain social ties with family and friends, and to engage with education or work.[22 25 26 32 35 43 66 67] Parents also anticipated difficulties balancing work and childcare obligations when they, or their children, needed to isolate.[22] These difficulties were described as major challenges to adherence to isolation[23] and stated as a reason for non-adherence to isolation guidance.[23 24 31 46]

People described guilt because their isolation would prevent them from performing their usual roles, meaning they were letting people down by not fulfilling their obligations and were placing an additional burden on work colleagues and household members.[26 51 66 68 69] Healthcare workers described their guilt in increasing the workload of already overworked colleagues when they needed to isolate.[70–72] This 'extra pressure on the remaining staff'[73] had organisational implications on staffing numbers and created an 'ethical dilemma' because of the impact on health service provision.[70 74]

> I'm a carer for vulnerable adults and would be worried about them receiving care that I could not give if I was isolating (Survey).[51]

### *Impact on mental health*

People also described negative effects of isolation on mental health,[57 68 75] including loneliness, boredom, lack of interaction with friends and family, the inability to exercise, strain on their interpersonal relationships and harmony within a household, and anxiety over finances.[26 29 32 34 35 43] University students in particular described struggling with losing access to their usual coping mechanisms[35] and that isolation exacerbated existing mental health problems.[29] Parents were concerned that isolating took school students away from their peers, education and outdoor activities, which risked their mental health.[43]

I think mental health was like my biggest challenge. It was very easy to just feel down and not wanting to do things, not feel motivated to either do work or just get out of bed (Focus group 8, student 5).[35]

The negative impact self-isolating had on mental health was described by others as a main barrier to self-isolation.[25 57] People anticipated that self-isolation would negatively impact their mental health,[35 51] and the unwillingness to self-isolate was exacerbated by people becoming 'fed up' with missing out on social interactions and other activities as the pandemic progressed.[52]

### Impact on finances

Existing financial hardship was described as an important barrier to isolating.[23 26 76] In addition, self-isolation created financial hardship that acted as a barrier to reporting positive test results, in order to avoid isolation and its financial consequences.[23 66 77–82] These challenges were in large part ascribed to loss of income through loss of work or wages.[25 51 52 56 75 77 83]

The consequences of self-isolation for certain groups in society, such as those who were homeless, misusing substances, or involved in the criminal justice system, may have differed from those experienced by the wider population, and circumstances may have made self-isolation more difficult to manage and influenced decisions to isolate for these groups.[32] In particular, the financial consequences and considerations were not uniform, with some reporting that they could absorb the loss of income or worked from home already, while others could not 'afford to isolate'.[18 75 79 83] The financial implications of isolation were described as particularly challenging for those with ethnic minority backgrounds, younger people, women, those on lower or precarious incomes, those unable to work from home while self-isolating, those working in 'shut-down sectors', migrant workers and those who were self-employed, impacting adherence to self-isolation recommendations in these groups.[19 34 78 83 84]

## DISCUSSION

This study describes six categories of factors that influenced the decision to self-isolate in the UK during the COVID-19 pandemic, mapped onto the COM-B framework as: capability (perceived ability and information and guidance), opportunity (logistics and social influences, including trust in authority) and motivation (perceived value and consequences).

### Decision to isolate

Ultimately, adherence to isolation guidance was a decision that was remade over the period of required isolation and as individuals' lives changed. Individuals' adherence to guidance ranged from full adherence at a personal cost,[20 85] to partial adherence where isolation was broken for reasons they believed to be 'essential' or 'acceptable',[22] to not isolating at all despite a positive test result[24 31] (including not disclosing their symptoms or

hiding their COVID-19 status[84]), to avoidance of testing in the first place.[84] This spectrum of engagement has been reported for other non-pharmaceutical measures across the COVID-19 and other pandemics, with people reporting a range of intentional and non-intentional non-adherence behaviours,[86] influenced at least in part by 'perceived necessity'.[2]

The decision to isolate was shaped by many factors.[22 24 35 57] These results highlight that people's decision to comply with isolation guidance was supported when the guidance was clear and plainly communicated, there were social norms to isolate and have trust in authority, there was support for planning and managing isolation, there was a perceived risk to themselves or others and there was clear perceived value of isolation in protecting others from COVID-19 transmission. However, practical logistical challenges, distrust of tests and the implementing institutions, and the negative consequences of isolating on daily life, mental health and finances all reduced adherence to isolation.

The factors influencing self-isolation adherence reflect the factors that influenced engagement with other COVID-19 public health measures. In the UK, clarity and consistency of guidance,[87 88] trust in authority,[88] perception of risk,[87] social norms[89] and 'collective importance'[90] were also found to influence adherence to other COVID-19 guidelines,[87] uptake of vaccination[90 91] and adherence to other non-pharmaceutical measures such as social distancing.[1 86 88 89] Mistrust in authority was found to explain a large proportion of the variance in vaccination hesitancy in the UK[90] and was repeatedly described in the context of non-adherence to social distancing regulations.[1 88] Adherence to different public health measures during COVID-19 was also linked; for example, those more hesitant to get vaccinated for COVID-19 were also less likely to adhere to social distancing guidelines.[90]

The desire to protect others was also a common theme facilitating engagement with many public health measures to tackle COVID-19[87 89 92] as well as previous pandemics.[2] Other behaviours required of the public during the COVID-19 response had some individual benefit: measures like vaccination or social distancing had personal benefit for the individual through reduction of the personal severity of disease or likelihood of contracting COVID-19, which increased engagement with vaccination programmes[93] or social distancing guidance.[92] Conversely, there was very little perceived direct individual benefit to self-isolation. Rather, the protection of others was commonly described as a major source of the value of isolation and as a 'fundamental driver of behaviour during the COVID-19 pandemic'.[45] Feelings of altruism resulted in moral obligations to engage in prosocial behaviours[94]; such behaviours are particularly important to curb transmission when uncomfortable and undesirable activities like isolation are required.[88]

The costs of self-isolation, however, were sufficiently high to disincentivise not only adherence to isolation guidance when testing positive, but also act as a deterrent

from testing for COVID-19 in the first place.[5] Unlike the safety concerns influencing uptake of COVID-19 vaccinations,[90] the concerns around the consequences of self-isolating were centred around the financial, practical and mental health effects of isolating. The interruption of daily life due to isolation was not only inconvenient[22 26 32 35 43 66 95] but also meant that obligations were not fulfilled, resulting in a transfer of the burden to family, friends and colleagues and creating feelings of guilt for the person isolating.[26 51 66 68 69] Isolation sometimes meant loss of work or even loss of a job; for many, this was too high a price to pay to prevent transmission,[25 51 75 77 83 85] echoing the impacts of the adverse socioeconomic effects of isolation that have made it a less acceptable non-pharmaceutical intervention across other pandemics.[2 96]

### Recommendations for future pandemic responses

Meaningful support has been described as a critical component of the strategy to encourage engagement with COVID-19 public health measures.[88] The major barrier to adherence to isolation guidance was the perception of personal consequences; therefore, external financial, practical and mental health support became an important facilitator of self-isolation.[16 22 25 32 33 76 79] Whether this came from the government, official agencies or informal networks, tangible support reduced the perception of isolation as a challenge.[23] However, support schemes did not consistently improve isolation adherence,[40 41] and uptake of the support offered was suboptimal.[30 37]

Not all individuals experienced the COVID-19 pandemic in the same way: for example, COVID-19 had a disproportionate effect on more vulnerable groups in the UK, with higher rates of infection, morbidity and mortality in some ethnic minorities, older people and those living in deprived areas.[97] Influential factors interact so that the motivation to self-isolate played out differently depending on people's capability and opportunity to isolate. While some described self-isolation easy to manage,[17] others found it unmanageable in various ways.[20 85] Contextual factors such as ethnicity and socio-economics played an important role in engagement with self-isolation, as well as with other programmes across different COVID-19 prevention measures,[86 98] for example, considerably reducing access to and uptake of vaccination[99 100] and increasing the negative impact of social distancing measures.[97] Therefore, support must be tailored to context and to an individual's specific challenges.[25] Advertising a range of easily accessible financial, practical and mental health support systems for different levels of requirement, based on individual circumstances as well as the intensity of isolation required, could help to streamline support services and make them more accessible.

Communication was described as a crucial tool to facilitate adherence to self-isolation recommendations.[39] Confusing or rapidly changing guidance was a repeated theme across behavioural research conducted during the COVID-19 pandemic in the UK and frequently reported as a barrier to engagement with public health programmes.[87] When guidance does not seem to make practical sense, it shifts the onus of decision-making onto individuals, who must make choices within the constraints of their personal priorities and context. Thus, official guidance must be particularly clear and unambiguous.[2] This could improve uptake across public health measures during a pandemic,[87] such as social distancing[1 97] or vaccination.[101]

Beyond clear, consistent guidance to support understanding of when, where and how to self-isolate[20 24 43–45] and providing information on available support,[30 36–38] it was repeatedly recommended that communication strategies should address people's motivation to comply[96] in order to activate personal norms that facilitate prosocial behaviours and engagement with preventative measures.[87 89 91] It was suggested that this could be achieved through messaging that reinforces trust in the government and programmes, demonstrates that adherence to guidance is the social norm, creates awareness of the consequences of non-adherence to guidance and highlights the perceived benefits to the wider community to appeal to people's altruism.[24 36 45 50 96]

Many of the factors influencing the decision to self-isolate may be weighed differently in another pandemic: the value of protecting others may change if a disease were to primarily affect different population groups, such as children, or if the severity differed from that of COVID-19. At the same time, if the negative consequences of isolating were mitigated, or the duration of required isolation was different, this would also shift the balance of the decision. Context matters in people's decision-making about health, and the decision to follow guidance cannot be considered in isolation.[102] Therefore, regardless of the lessons learnt during the COVID-19 pandemic and their applicability to future pandemic preparedness, research is needed to understand what affects how people weigh the different factors influencing the decision to isolate and what could be leveraged to encourage people to prioritise the value over the costs in future health emergencies. Additionally, as perceptions and decisions changed over time, measures should be in place to collect real-time data in future pandemics and inform decisions.

### Strengths and limitations

Most evidence included in this review was concerned with factors influencing the decision to test[5] and thus evidence on isolation behaviour was often drawn from studies with a primary focus on testing during COVID-19. Additionally, this review was conducted in a short space of time with a large team, so multiple people were involved in screening and extracting data, which could have introduced selection bias. Steps to mitigate this included overlap of screening and data extraction for a proportion of the sources, quality checks and review of original documents during write up. However, it is strengthened by the

broad search strategy that comprehensively scoped the available literature, as well as by the inclusion of a large body of unpublished and confidential internal data from UKHSA. These data from UKHSA were able to supplement the paucity of information available on the factors affecting isolation behaviour in the published literature, and would otherwise not be available for synthesis, general dissemination and contribution to discussions on future strategies. Additionally, we consulted a diverse set of stakeholders to identify additional sources of evidence and to contextualise and interpret the results as accurately as possible.

## Conclusions

This review demonstrates that the decision to isolate was a complex and dynamic process influenced by multiple factors. All dimensions of the COM-B behavioural model must be in place for people to comply with isolation guidance: in addition to leveraging the value in protecting others and mitigating the negative consequences to motivate people to self-isolate, future pandemic response policies and programmes will need to ensure that individuals have the capability and opportunity to put that motivation into practice. Communication of guidelines must be clear and consistent; context-appropriate financial, practical and mental health support is required; and building trust with the public will be critical to ensure the success of any public health response to a pandemic.

**Acknowledgements** We would like to extend our gratitude to the UK Health Security Agency for their invaluable collaboration in this scoping review. Their willingness to provide access to confidential internal documents significantly enriched our research and understanding of isolation behaviours during the COVID-19 pandemic in England. We would like to thank by name the team who helped to organise, screen and extract the data: Ainura Moldokmatova, Firdaus Mohandas, Katie Douglass, Umar Mahmood and Zoe Echanah. We would also like to thank Adam Bodley for his contribution to the editing process. We would like to acknowledge the contributions of the scientific, clinical and technical experts who have offered valuable input, advice and support throughout the evaluation and preparation of the overarching report, particularly the contributions of the Scientific Advisory Group. Our thanks also go to the numerous healthcare professionals, policy-makers and community leaders who worked tirelessly through COVID-19 pandemic in England to ensure that patients received quality care and that communities were safe.

**Collaborators** Ainura Moldokmatova (Big Data Institute, Li Ka Shing Centre for Health Information and Discovery, Nuffield Department of Medicine, University of Oxford); Anastasiia Polner (EY Seren, Ernst & Young LLP London, UK); Angus Ferguson-Lewis (EY Seren, Ernst & Young LLP London, UK); Ben Lambert (College of Engineering, Mathematics and Physical Sciences, University of Exeter, Exeter, UK, Department of Statistics, University of Oxford); Billie Andersen-Waine (EY Seren, Ernst & Young LLP London, UK); Bo Gao (Centre for Tropical Medicine and Global Health, Nuffield Department of Medicine, University of Oxford); Caroline Franco (Nuffield Department of Clinical Medicine, University of Oxford | Institute of Theoretical Physics - São Paulo State University (UNESP)); Claire Keene (NDM Centre for Global Health Research, Nuffield Department of Medicine, University of Oxford); Emily Rowe (Ernst & Young(EY) UKI Health Sciences and Wellness, London, UK); Jared Norman (Modelling and Simulation Hub, Africa (MASHA), University of Cape Town); Kasia Stepniewska (Infectious Diseases Data Observatory, Nuffield Department of Medicine, University of Oxford, UK); Kweku Bimpong (Ernst & Young(EY) UKI Health Sciences and Wellness, London, UK); Liberty Cantrell (Department of Paediatrics, University of Oxford); Lisa J White (Big Data Institute, Li Ka Shing Centre for Health Information and Discovery, Nuffield Department of Medicine, University of Oxford, Oxford, UK); Joseph L-H Tsui (Department of Biology, University of Oxford); Ma'ayan Amswych (Ernst & Young(EY) UKI Health Sciences and Wellness, London, UK); Marta Wanat (Nuffield Department of Primary Care, University of Oxford); Melinda C Mills (Leverhulme Centre for Demographic Science (LCDS), University of Oxford); Merryn Voysey (Oxford Vaccine Group, Department of Paediatrics, University of Oxford, Oxford, UK); Muhammad Kasim (Department of Physics, University of Oxford); Prabin Dahal (Infectious Diseases Data Observatory, Nuffield Department of Medicine, University of Oxford, UK); Rachel Hounsell (Big Data Institute, Li Ka Shing Centre for Health Information and Discovery, Nuffield Department of Medicine, University of Oxford); Reshania Naidoo (Ernst & Young(EY) UKI Health Sciences and Wellness, London, UK NDM Centre for Global Health Research, Nuffield Department of Medicine, University of Oxford); Richard Lewis (Richard Lewis Consulting Ltd.); Rima Shretta (NDM Centre for Global Health Research, Nuffield Department of Medicine, University of Oxford); Randolph Ngwafor Anye (Nuffield Department of Primary Care Health Sciences, University of Oxford); Ricardo Aguas (Centre for Tropical Medicine and Global Health, Nuffield Department of Medicine, University of Oxford); Richard Creswell (Department of Computer Science, University of Oxford); Sabine Dittrich (Centre for Tropical Medicine and Global Health, Nuffield Department of Medicine, University of Oxford); Sassy Molyneux (Centre for Tropical Medicine and Global Health, Nuffield Department of Medicine, University of Oxford); Siyu Chen (Centre for Tropical Medicine and Global Health, Nuffield Department of Medicine, University of Oxford); Sheetal Silal (Centre for Tropical Medicine and Global Health, Nuffield Department of Medicine, University of Oxford, MASHA, University of Cape Town); Sompob Saralamba (Centre for Tropical Medicine and Global Health, Nuffield Department of Medicine, University of Oxford); Sophie Dickinson (Ernst & Young(EY) UKI Health Sciences and Wellness, London, UK); Sumali Bajaj (Department of Biology, University of Oxford); Sunil Pokharel (Centre for Tropical Medicine and Global Health, Nuffield Department of Medicine, University of Oxford); Tracy Evans (Nuffield Department of Primary Care Health Sciences, University of Oxford); Umar Mahmood (Ernst & Young(EY) UKI Health Sciences and Wellness, London, UK); Wirichada Pan-ngum (Centre for Tropical Medicine and Global Health, Nuffield Department of Medicine, University of Oxford).

**Contributors** Conceptualisation and methodology: CMK, BA-W, SM, MW, RN and LW. Analysis, integration and interpretation: CMK, SD, BA-W, AF-L, AP, MA, SM, MW and the members of the EY-Oxford Health Analytics Consortium. Writing – original draft: CMK and SD. Writing – review, editing and approval: RN, BA-W, AF-L, AP, MA, LW, SM, MW and the members of the EY-Oxford Health Analytics Consortium. All named authors read and approved the final manuscript. Guarantors: CK and MW.

**Funding** This study was funded by Secretary of State for Health and Social Care acting as part of the Crown through the UK Health Security Agency (UKHSA), reference number C80260/PRO5331.

**Competing interests** All authors had financial support from the UK Health Security Agency (UKHSA) for the submitted work; EY LLP London has previously received payment for consultancy work and advisory on the NHS Test & Trace response from the UK Department of Health and Social Care, now known as the UK Health Security Agency. There are no other relationships or activities that could appear to have influenced the submitted work.

**Patient and public involvement** Patients and/or the public were not involved in the design, or conduct, or reporting, or dissemination plans of this research.

**Patient consent for publication** Not applicable.

**Ethics approval** Not applicable.

**Provenance and peer review** Not commissioned; externally peer reviewed.

**Data availability statement** Data are available upon reasonable request. Data are included as supplementary material. Further data requests can be directed to the corresponding author.

**ORCID iDs**
Claire Marriott Keene http://orcid.org/0000-0002-0875-5884
Reshania Naidoo http://orcid.org/0000-0002-3368-0255
Marta Wanat http://orcid.org/0000-0002-0163-1547

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
