## [Reviewer comments · BMJ Open]

ARTICLE DETAILS

TITLE (PROVISIONAL)	Decision to self-isolate during the COVID-19 pandemic in the UK: a rapid scoping review
AUTHORS	Keene, Claire; Dickinson, Sophie; Naidoo, Reshania; Andersen-Waine, Billie; Ferguson-Lewis, Angus; Polner, Anastasia; Amswych, Ma'ayan; White, Lisa; Molyneux, Sassy; Wanat, Marta; EY-Oxford Health Analytics Consortium, NA

VERSION 1 – REVIEW

REVIEWER	Seale, Holly University of New South Wales, School of Public Health and Community Medicine
REVIEW RETURNED	29-Jan-2024

GENERAL COMMENTS	1. Could you add a short breakdown of the main reasons for papers not being included in the review.2. Did any of the reports focus on particular populations of risk? i.e. Homeless populations, those who identify as ethnic minority, elderly? In terms of the first theme around perceived ability to self-isolate, what were the factors/characteristics amongst those people who reported the difficulties in isolating.
--

VERSION 1 – AUTHOR RESPONSE

Reviewer: 1

Dr. Holly Seale, University of New South Wales

Comments to the Author:

1. Could you add a short breakdown of the main reasons for papers not being included in the review. We have included this in the PRISMA ScR flow diagram, and have added a sentence to signpost to it in the 'Overview of the evidence' section in the results.

2. Did any of the reports focus on particular populations of risk? i.e. Homeless populations, those who identify as ethnic minority, elderly? In terms of the first theme around perceived ability to self-isolate, what were the factors/characteristics amongst those people who reported the difficulties in isolating. Indeed some of the reports did focus on specific populations, and the perceptions and experiences of these groups are included in this manuscript. We have included a paragraph in the discussion to explore the difference in experiences and perceptions between different populations, and we have edited this to add clarity in response to this comment. However, we focused on people's experiences and perceptions of self-isolating using qualitative data, and as we did not include any quantitative analyses to examine the association of characteristics with the outcome of self-isolation, or the perception of difficulty isolating, we are unable to answer this reviewer question with our data.